# Dirac Hydrodynamics in 19 Forms

**Luca Fabbri** [1,2]

1 DIME, Sez. Metodi e Modelli Matematici, Università di Genova, Via all'Opera Pia 15, 16145 Genova, Italy; luca.fabbri@edu.unige.it
2 Istituto Nazionale Fisica Nucleare, Sezione di Genova, Via Dodecaneso 33, 16146 Genova, Italy

**Abstract:** We consider the relativistic spinor field theory re-formulated in polar variables to allow for the interpretation given in terms of fluid variables. After that, the dynamics of spinor fields are converted as dynamics of a special type of spin fluid. We demonstrate that such conversion into dynamical spin fluid is not unique, but it can be obtained through 19 different rearrangements, by explicitly showing the 19 minimal systems of hydrodynamic equations that are equivalent to the Dirac equations.

**Keywords:** spinor field; polar form; hydrodynamics

## 1. Introduction

In mathematics, writing complex functions as a product of a module times a unitary phase, that is, writing a complex function in its polar decomposition, is a well-known method extensively used to treat a variety of problems.

In physics, this method can be applied whenever complex functions are involved, and of all cases, the most important is certainly quantum mechanics. In its non-relativistic spinless case, the wave function is a complex scalar, so its polar decomposition is straightforward. When the wave function written in the polar form is inserted into the Schrödinger equation, the latter splits in a continuity equation for the velocity density plus a quantum mechanical extension of the Hamilton–Jacobi equation. This procedure was first applied by Madelung, and it is widely known today. As in their final form the field equations are expressed as derivatives of quantities like density and velocity, this procedure actually converts quantum mechanics into a special type of fluid mechanics, and since the continuity equation states that this fluid must be incompressible, such a formulation of quantum mechanics is called the hydrodynamic formulation [1,2].

A more realistic description of quantum mechanics can be given when it is extended to include spin. Still within the validity of non-relativistic regimes, the wave function is now a complex doublet that transforms in a specific way under rotations, but a polar form can be obtained just the same. After the inclusion of spin, the Schrödinger equation is known to be enlarged into the Pauli equations, but again, when the spinorial wave function written in the polar form is used, a hydrodynamic formulation becomes possible [3,4]. As it stands, one would now assume that in the relativistic case, in which the wave function is formed by two complex doublets transforming in a very specific way under boosts and rotations, an analogous polar form could also be obtained. And since in the relativistic case the Pauli equations are replaced by the Dirac equations, one should also expect the relativistic spinorial field in the polar form to convert the Dirac equations into the corresponding hydrodynamic formulation [5,6]. This is exactly what eventually happens although not with the immediate chronology that one might have guessed. And the reason is covariance.

In fact, in the passage from a wave function of a scalar character to that of a spinor character, whether non-relativistic or relativistic, there are additional transformation laws that have to be considered. As already mentioned, non-relativistic spinors are doublets

that account for both helicity states, and while each component is a scalar for diffeomorphisms (that is, for passive transformations of coordinates), the two components mix under rotations (that is, for active changes in frame). Richer in structure, relativistic spinors are columns of four components that account for both chiral states as well as both helicity states, and while, again, each component is a scalar for diffeomorphisms, the four components mix under Lorentz transformations. It is now easier to see where the difficulty may be found. Indeed, considering the relativistic spinor, the passage to its polar form would be implemented when, for the four complex components, each one is written as a product of a module times a phase. In general, therefore, there are four different modules as well as four different phases, all mixing between each other under Lorentz transformations [7].

Such a difficulty can be found ubiquitously in the literature, not only in the works about the polar form, but more in general in all investigations that aim at studying spinors in terms of tensor quantities. The idea of writing a spinor in terms of tensor quantities dates back to Cartan himself, and it continued immediately after him with the works of Whittaker [8], Ruse [9] and Taub [10]. In the same years, Yvon [11], and later Takabayasi [12,13] also contributed to this enterprise. And, in more recent times, new elements were brought into the game by Hestenes [14–18] in terms of space-time algebra. For a very comprehensive and clear account of all these results, and a more extensively detailed source of references, we address the interested reader to the very recent book of Zhelnorovich [19].

In any case, there remains the mentioned issue of manifest covariance in general cases. While the cumulative results clearly support the idea that spinors, in spite of being complex objects, can nevertheless be written in terms of real tensors, these approaches are either tackling the problem component by component, thus lacking the manifest form of covariance [8–11], or treating it in a manifest covariant way, although restricted to specially relativistic situations [12–18] or performing the study for generally relativistic cases in curved space-times, but always in a preferred basis [19]. Neither in these works, nor anywhere else in the literature, a manifestly covariant general study is found.

In this quest for generality, the attempt that proceeded the farthest, to the best of our knowledge, is that of Jakobi and Lochak in [20,21], where the spinor field was written in terms of real tensors by means of the polar decomposition without the use of any preferred basis, although, again, we have no knowledge of any attempts to investigate such polar decomposition at a differential level. For what we can tell, it is only very recently that, upon introduction of suitable objects called tensorial connections, it has become possible to write the polar form of the covariant derivative of the spinor field in such a way that its structure is manifestly covariant under general transformations [22].

Correspondingly, it has finally become possible to write the hydrodynamic form of the Dirac equations in such a way that its structure is manifestly covariant under general transformations, in space-times that are curved, with torsion, and in the presence of electrodynamics [23]. In this paper, we ask whether this hydrodynamic form is unique, and we demonstrate that this is not so. We prove that there are, in fact, 19 different reconfigurations of the hydrodynamic form of the Dirac equations, and that such 19 systems of field equations are the minimal ones that are equivalent to one another, in the sense that from any one of such systems there is not a single field equation that can be removed without also producing the loss of equivalence to the original Dirac equations.

## 2. Dirac Hydrodynamics

### 2.1. Kinematic Quantities

2.1.1. Spinor Fields in the Polar Form

We start this first section by recalling the general ideas of the conversion of the Dirac theory in its hydrodynamic formulation. As stated in the introduction, for us, Dirac hydrodynamics, or relativistic spinning quantum mechanics in hydrodynamic form, is just the relativistic extension with a spin of quantum mechanics in hydrodynamic form as it was initially conceived by Madelung. As such, it is also taken as a synonym to spinor theory in the polar form.

We begin the presentation by recalling the notations. First of all, we define the Clifford matrices $\gamma^a$ as the set of matrices verifying $\{\gamma_a, \gamma_b\} = 2\mathbb{I}\eta_{ab}$, where $\eta_{ab}$ is the Minkowski matrix. From them, we define $[\gamma_a, \gamma_b]/4 = \sigma_{ab}$ verifying $2i\sigma_{ab} = \varepsilon_{abcd}\pi\sigma^{cd}$ where $\varepsilon_{abcd}$ is the completely antisymmetric pseudo-tensor and in which the $\pi$ matrix is defined. This matrix is what is traditionally indicated by $\gamma^5$ as the fifth matrix after $\gamma^1$, $\gamma^2$, $\gamma^3$, $\gamma^4$ in Kaluza–Klein theories, at the time when the temporal coordinate was still designated as the fourth coordinate. Nowadays, we use a zero to indicate the temporal gamma matrix, and thus we should use a four for the fifth gamma matrix, or better still, we should just drop an index that no longer corresponds to any actual dimension. But even worse, in some signatures, the position of the index five, whether upper or lower, changes the sign of the gamma matrix, with ensuing risks of errors that become possible. To avoid the potential for mistakes, we prefer to employ a notation in which there is no index at all. The choice of $\pi$ instead of $\gamma$ is explained by the fact that the matrix is parity-odd as opposed to the usual gammas being parity-even. This is not dissimilar to the normally accepted choice of using $\sigma$ for the commutators of gamma matrices. Identity $\gamma_i\gamma_j\gamma_k = \gamma_i\eta_{jk} - \gamma_j\eta_{ik} + \gamma_k\eta_{ij} + i\varepsilon_{ijkq}\pi\gamma^q$ shows that any product of more than two matrices can always be reduced to two matrices, therefore suggesting that the set $(\mathbb{I}, \gamma^a, \sigma_{ab}, \gamma^a\pi, \pi)$ is complete, and so it is also a basis for the space of $4 \times 4$ complex matrices. From these definitions, it is straightforward that $\sigma_{ab}$ are the generators of the Lorentz algebra. Exponentiation of the generators offers $\mathbf{\Lambda}$ as the element of the Lorentz group. We notice that $\mathbf{\Lambda}$ is a complex Lorentz transformation, as opposed to $\Lambda^a_b$ being the real Lorentz transformation. The two are linked by the relation $\mathbf{\Lambda}\gamma^b\mathbf{\Lambda}^{-1}\Lambda^a_b = \gamma^a$ which, in turn, also specifies that the Clifford matrices are all constant, as is expected and well known. With the complex Lorentz transformation $\mathbf{\Lambda}$ and a general unitary phase $e^{iq\alpha}$, we define $S = \mathbf{\Lambda}e^{iq\alpha}$ as the spinorial transformation. The spinorial transformation is therefore just the product of boosts and rotations as well as a gauge shift of charge $q$ as it is supposed to be to account for both space-time and electrodynamic transformations.

So far, nothing is new, and, aside from the specific convention we employ, all can be found in textbooks. With these tools, we define spinor fields as objects that, under spinorial transformations, transform according to

$$\psi \rightarrow S\psi \qquad \text{and} \qquad \overline{\psi} \rightarrow \overline{\psi}S^{-1}, \tag{1}$$

where $\overline{\psi} = \psi^\dagger\gamma^0$ is the (unique) adjunction procedure. With these adjoint spinors, we can construct the bi-linears,

$$\Sigma^{ab} = 2\overline{\psi}\sigma^{ab}\pi\psi \qquad M^{ab} = 2i\overline{\psi}\sigma^{ab}\psi \tag{2}$$

$$S^a = \overline{\psi}\gamma^a\pi\psi \qquad U^a = \overline{\psi}\gamma^a\psi \tag{3}$$

$$\Theta = i\overline{\psi}\pi\psi \qquad \Phi = \overline{\psi}\psi \tag{4}$$

which are all real tensors. We defined six of them for reasons of symmetry and simplicity, but as it is clear from the linear independence of the Clifford matrices, they are not all linearly independent. In fact,

$$\Sigma^{ij} = -\tfrac{1}{2}\varepsilon^{abij}M_{ab}, \tag{5}$$

showing that the two antisymmetric tensors are just the Hodge dual of one another. By taking only one of them, $M_{ab}$, for example, together with the other four bi-linears, we may form a set of bi-linears that are all linearly independent, although not independent. Indeed,

$$M_{ab}(\Phi^2 + \Theta^2) = \Phi U^j S^k \varepsilon_{jkab} + \Theta U_{[a}S_{b]}, \tag{6}$$

showing that if $\Phi^2 + \Theta^2 \neq 0$; then, $M_{ab}$ can also be dropped in favour of the two vectors and the two scalars. The axial vector and the vector with the pseudo-scalar and scalar are also not independent since

$$U_a U^a = -S_a S^a = \Theta^2 + \Phi^2, \tag{7}$$

$$U_a S^a = 0, \tag{8}$$

and, in the case in which $\Phi^2 + \Theta^2 \neq 0$, we can see that the axial vector is space-like, while the vector is time-like. Finally,

$$2\sigma^{\mu\nu} U_\mu S_\nu \boldsymbol{\pi}\psi + U^2\psi = 0, \tag{9}$$

$$i\Theta S_\mu \gamma^\mu \psi + \Phi S_\mu \gamma^\mu \boldsymbol{\pi}\psi + U^2\psi = 0, \tag{10}$$

to complete the list of identities between the bi-linears that we use below.

Now, a result that can be proven is that under the general assumption $\Phi^2 + \Theta^2 \neq 0$, it is always possible to write any spinor field in the polar form, which in a chiral representation is given according to

$$\psi = \phi\, e^{-\frac{i}{2}\beta\boldsymbol{\pi}}\, \boldsymbol{L}^{-1} \begin{pmatrix} 1 \\ 0 \\ 1 \\ 0 \end{pmatrix} \tag{11}$$

for a pair of functions $\phi$ and $\beta$ and for some $\boldsymbol{L}$ having the structure of a spinorial transformation and such that the polar form is unique up to the discrete transformation $\beta \rightarrow \beta + \pi$ and up to the reversal of the third axis [20,21]. The three elements $\boldsymbol{L}$, $\phi$ and $\beta$ in (11) need some explanation. After that, (11) is substituted in (4); one obtains

$$\Theta = 2\phi^2 \sin\beta, \qquad \Phi = 2\phi^2 \cos\beta, \tag{12}$$

showing that $\phi$ and $\beta$ are a real scalar and a real pseudo-scalar called the module and the chiral angle. Equipped with the polar form, we can also normalize

$$S^a = 2\phi^2 s^a, \qquad U^a = 2\phi^2 u^a, \tag{13}$$

where $u^a$ and $s^a$ are the velocity vector and the spin axial vector. Identities (7) and (8) reduce to

$$u_a u^a = -s_a s^a = 1, \tag{14}$$

$$u_a s^a = 0, \tag{15}$$

showing that the velocity has only three independent components given by the three components of its spatial part (indeed, constraint $u_a u^a = 1$ fixes the temporal component), whereas the spin has only two independent components, assigned by the two angles that, in the rest frame, its spatial part forms with the third axis (in fact, because of constraint $u_a s^a = 0$, the temporal component of the spin is zero in the rest frame, and due to constraint $s_a s^a = -1$, one of the spatial components is fixed). The physical interpretation of this mathematical fact is that, in the frame at rest, there remains no spatial component of the velocity and its temporal component is unity, and because this also implies the vanishing of the temporal component of the spin, its spatial component, when aligned along the third axis, selects this axis as the axis of symmetry of the system. Hence, any rotation around this axis would have to be an irrelevant rotation, as it would be unable to have any effect on the spinor itself. In the following sections, we still refer to rotations around the third axis because we chose this axis for all explicit computations, although it is clear that

from a covariant perspective, we would refer to them as rotations around the spin axis. The remaining identities (9) and (10) are

$$2\sigma^{\mu\nu}u_\mu s_\nu \boldsymbol{\pi}\psi + \psi = 0, \tag{16}$$

$$is_\mu \gamma^\mu \psi \sin\beta + s_\mu \gamma^\mu \boldsymbol{\pi}\psi \cos\beta + \psi = 0, \tag{17}$$

which are useful later on. As for $L$, we know that it has the structure of a spinorial transformation, and thus it is a product of a gauge times a Lorentz transformation. Its interpretation is of straightforward reading. It is the specific transformation that takes an assigned spinor into its rest frame with the spin aligned along the third axis, however general the initial spinor is. As such, $L$ should not be confused with $S$ because even if mathematically they are the same and both are a spinorial transformation, from a physical perspective, they are very different, with $S$ indicating the way in which spinors behave under transformations and $L$ indicating how any given spinor can always be seen as a suitable deformation of its simplest rest frame spin-eigenstate form. Metaphorically, if the spinor were to be a top spinning on a table, then $S$ would indicate how to move from the fixed system of reference in which the table is at rest to the rotating system of reference in which the top is at rest while $L$ would indicate how the top is spinning. Therefore, the advantage of writing spinor fields in the polar form is that in their four complex components, the eight real functions are re-organized in such a way that the two real scalars, that is, the two true degrees of freedom ($\phi$ and $\beta$), remain isolated from the six real parameters, which can always be transferred into the frame (encoded within $L$). The fact that $L$ is the product of a gauge times a Lorentz transformation might in principle lead us to think that it has, in total, $1 + 6 = 7$ parameters, and this appears to be in contradiction with the fact that it effectively has only six parameters as we have just explained. The resolution of this paradox is that, as is clear from (11), the action of a gauge transformation is indistinguishable from the action of a rotation around the third axis. The consequence is that there is a redundancy between the phase and the angle of rotation around the third axis, so that the a priori seven parameters are in fact reduced to six parameters alone. If the total parameters of the combined gauge and Lorentz transformations are essentially six and one is encoded by the gauge transformation, then there can be no more than five parameters that remain in the Lorentz transformation, and this is precisely what happens. The five parameters are given by the three rapidities of the velocity vector and the two Euler angles of the spin axial vector, as we discussed above. This redundancy should not be surprising, and, indeed, it is rather common. An identical situation happens in the Standard Model, where the $U(1) \times SU(2)$ group has four parameters, but the hypercharge and the third component of the isospin combine to form a single parameter, so that the entire group does not have four but three Goldstone bosons solely. We notice that this is more than an example; it is a true mathematical analogy. Indeed, the fact that the parameters of $L$ can always be transferred into the frame means that they are the Goldstone bosons associated to the spinor field by definition. The Goldstone bosons we have here play for the spinor field exactly the same role that the Goldstone bosons in the Standard Model play for the Higgs field as demonstrated in reference [22]. We return back to this after discussing what happens at the differential level.

For now, we notice that the preferred frame mentioned in [19] is the frame in which $L = \mathbb{I}$ with the spinor at rest and in the spin eigenstate. The generality achieved in [20,21] is hence due to the $L$ operator.

### 2.1.2. Tensorial Connections

For general spinor fields, the spinorial covariant derivative is defined according to

$$\nabla_\mu \psi = \partial_\mu \psi + \boldsymbol{C}_\mu \psi \tag{18}$$

in terms of the spinorial connection $\boldsymbol{C}_\mu$, which is itself defined by its transformation law

$$\boldsymbol{C}_\mu \to S(\boldsymbol{C}_\mu - S^{-1}\partial_\mu S)S^{-1}, \tag{19}$$

where $\boldsymbol{S}$ is the spinorial transformation law. This spinorial connection can be decomposed according to

$$\boldsymbol{C}_\mu = \tfrac{1}{2} C^{ab}{}_\mu \boldsymbol{\sigma}_{ab} + iqA_\mu \mathbb{I},\tag{20}$$

where $C^{ab}{}_\mu$ is the spin connection of the tetrads of the space-time and $A_\mu$ is the gauge potential. We do not provide, in the present work, the details of the explicit form of the spin connection written in terms of the tetrad fields because this is very standard material of any introductory course of General Relativity in the tetradic formalism [19].

Now, when in the spinorial covariant derivative the spinor field is written in the polar form, something interesting starts to emerge. To see what, recall that one can always write

$$\boldsymbol{L}^{-1}\partial_\mu \boldsymbol{L} = iq\partial_\mu\xi\mathbb{I} + \tfrac{1}{2}\partial_\mu\xi_{ij}\boldsymbol{\sigma}^{ij}\tag{21}$$

for some $\partial_\mu\xi$ and $\partial_\mu\xi_{ij}$ which are in fact the Goldstone fields of the spinor. In (21), the redundancy between phase and angle of rotation around the third axis shows that the apparent seven Goldstone fields are effectively six Goldstone fields only, since $\partial_\mu\xi_{12}$ can always be re-absorbed in a redefinition of $q\partial_\mu\xi$ in general (in fact, component $\partial_\mu\xi_{12}$ multiplies $\sigma^{12}$ while $q\partial_\mu\xi$ multiplies the identity matrix, and because for rest frame spin-eigenstates the action of $\sigma^{12}$ is equivalent to that of the identity matrix by definition, we have components $\partial_\mu\xi_{12}$ and $q\partial_\mu\xi$ always displaying some redundancy, as we also discussed before). In $q\partial_\mu\xi$, the presence of the charge has been made explicit so as to simplify the definition of quantities

$$q(\partial_\mu\xi - A_\mu) \equiv P_\mu,\tag{22}$$

$$\partial_\mu\xi_{ij} - C_{ij\mu} \equiv R_{ij\mu},\tag{23}$$

since now $q$ can be collected in the definition of $P_\mu$ in (22). It is now possible to see that, after that the Goldstone fields are transferred into the frame, they combine with gauge potential and spin connection to become the longitudinal components of the $P_\mu$ and $R_{ij\mu}$ objects, which then turn into a real vector and a real tensor called gauge and space-time tensorial connections, as it has been demonstrated in [22]. With (22) and (23), we finally have

$$\nabla_\mu\psi = \left(-\tfrac{i}{2}\nabla_\mu\beta\boldsymbol{\pi} + \nabla_\mu\ln\phi\mathbb{I} - iP_\mu\mathbb{I} - \tfrac{1}{2}R_{ij\mu}\boldsymbol{\sigma}^{ij}\right)\psi\tag{24}$$

as the polar form of the spinor field covariant derivative. From it, we can deduce that

$$\nabla_\mu s_i = R_{ji\mu}s^j \qquad \nabla_\mu u_i = R_{ji\mu}u^j\tag{25}$$

are valid as general identities. Recall that rotations around the spin axis are unable to have effects on any component of $s_i$ and $u_i$ and therefore on $\nabla_\nu s_i$ and $\nabla_\nu u_i$ in general. Because of (25), this means that the components of $R_{ab\nu}$ that correspond to the rotations around the spin axis remain undetermined. By using (25), we can write

$$\begin{aligned}R_{ab\mu} \equiv {}&R_{ab\mu} - u_a\nabla_\mu u_b + u_b\nabla_\mu u_a - s_b\nabla_\mu s_a + s_a\nabla_\mu s_b - (u_a s_b - u_b s_a)\nabla_\mu u_k s^k +\\&+u_a\nabla_\mu u_b - u_b\nabla_\mu u_a + s_b\nabla_\mu s_a - s_a\nabla_\mu s_b + (u_a s_b - u_b s_a)\nabla_\mu u_k s^k \equiv\\\equiv {}&u_a\nabla_\mu u_b - u_b\nabla_\mu u_a + s_b\nabla_\mu s_a - s_a\nabla_\mu s_b + (u_a s_b - u_b s_a)\nabla_\mu u_k s^k +\\&+\tfrac{1}{2}R_{ij\mu}\varepsilon^{ijcd}\varepsilon_{abpq}s_c u_d s^p u^q,\end{aligned}\tag{26}$$

and therefore one can always write

$$R_{ab\mu} = u_a\nabla_\mu u_b - u_b\nabla_\mu u_a + s_b\nabla_\mu s_a - s_a\nabla_\mu s_b + (u_a s_b - u_b s_a)\nabla_\mu u_k s^k + 2\varepsilon_{abij}u^i s^j V_\mu\tag{27}$$

for some vector $V_\mu = \tfrac{1}{4}R_{ij\mu}\varepsilon^{ijcd}u_c s_d$ that is no more specified. This vector is precisely what represents the components of $R_{ab\nu}$ that correspond to rotations around the spin axis, as

is clear from the fact that $2V_\mu = R_{12\mu}$ whenever the spinor field is in its rest frame with the spin aligned along the third axis. As we commented in the previous sub-section, a general spinor field can always be seen as the simplest spinor field at rest and with the spin aligned along the third axis after a suitable deformation, and as we see now, such a deformation $L$ has the structure of a spinorial transformation in which generator $ij$ has a parameter whose derivative with respect to the $\mu$th coordinate is given by component $R_{ij\mu}$ of the tensorial connection. As deformation $L$ can be interpreted like a strain, tensorial connection $R_{ij\mu}$ is interpretable like the strain-rate tensor. This analogy is very strong if we consider that $R_{ij\mu}$ in (27) can be written in terms of derivatives of spin and velocity, and that the derivative of velocity is the same object with which the strain rate tensor is constructed in continuum mechanics. Together, gauge and space-time tensorial connections $R_{ij\mu}$ and $P_\mu$ have a total of $24 + 4 = 28$ components, so that once we subtract the four components that cannot be determined, we remain with an effective total of 24 components. These 24 components, added to the $4 \times 2 = 8$ components of $\nabla_\mu \phi$ and $\nabla_\mu \beta$, make up for the full 32 components of the spinor field covariant derivative. The full counting of components is perfectly matched. As for the parallel with the Standard Model that we discussed in the previous sub-section, we can say that the tensorial connections $R_{ij\mu}$ and $P_\mu$ are just the geometric and electrodynamic analog of vector bosons $W_\nu^\pm$ and $Z_\nu$ and the frame in which the spinor field is at rest and with spin along the third axis is the analog of the unitary gauge [22]. As we said, these analogies are due to a strict mathematical parallel between the polar form of spinor fields and the Higgs field in the Standard Model, a parallel holding up until symmetry breaking.

To complete the comparison with previous works, we can finally say that it is precisely the definition of the tensorial connection that allowed the results of Jakobi and Lochak [20,21] on the polar form of the spinor field to be extended to its differential structures [22]. And from this, we can now move to the dynamics.

### 2.2. Dynamical Equations

In the previous sub-section, we introduced the differential structures, the gauge and space-time tensorial connections, with which to define the covariant derivative of spinor fields in the polar form. Readers may have noticed that everything has been performed by taking into account the tetradic structure, and therefore the metric structure, of the space-time itself. Nevertheless, we neglected all torsional degrees of freedom. A reader that cares about differential geometry in its most general form might now complain that torsion should be allowed instead. In order for this to be achieved, one can simply notice that, because torsion is a true tensor, it is always possible to decompose the most general connection into the torsionless connection plus the torsional contributions. Therefore, the most general differential geometry with torsion is always equivalent to the differential geometry with no torsion plus an additional field representing the torsion tensor. As a consequence, full generality can be restored in the dynamics by the addition of the torsion field. Because torsion couples to spin, which, in the case of Dirac spinors, is completely antisymmetric, torsion has a completely antisymmetric part only, which is equivalent to the Hodge dual of an axial vector. Hence, in the dynamics, we add torsion in the form of an axial vector field. The reader interested in more details can find them in [24].

Therefore, the dynamics of the spinor field are given here in terms of Dirac equations,

$$i\gamma^\mu \nabla_\mu \psi - XW_\mu \gamma^\mu \boldsymbol{\pi} \psi - m\psi = 0, \tag{28}$$

with $W_\mu$ being the axial vector torsion and $X$ being the coupling constant of the torsion–spin interaction. Now, by multiplying (28) on the left with $\mathbb{I}$, or $\gamma^a$, or $\sigma_{ab}$, or $\gamma^a \boldsymbol{\pi}$, or $\boldsymbol{\pi}$, and then,

in each case, also with $\overline{\psi}$, we obtain five equations that, after splitting real and imaginary parts, produce ten equations that are all real and tensorial in structure. They are given by

$$\nabla_\mu U^\mu = 0, \tag{29}$$

$$\frac{i}{2}(\overline{\psi}\gamma^\mu \boldsymbol{\pi}\nabla_\mu\psi - \nabla_\mu\overline{\psi}\gamma^\mu \boldsymbol{\pi}\psi) - XW_\sigma U^\sigma = 0, \tag{30}$$

$$\nabla^{[\alpha}U^{\nu]} + i\varepsilon^{\alpha\nu\mu\rho}(\overline{\psi}\gamma_\rho\boldsymbol{\pi}\nabla_\mu\psi - \nabla_\mu\overline{\psi}\gamma_\rho\boldsymbol{\pi}\psi) - 2XW_\sigma U_\rho\varepsilon^{\alpha\nu\sigma\rho} - 2mM^{\alpha\nu} = 0; \tag{31}$$

$$\nabla_\mu S^\mu - 2m\Theta = 0, \tag{32}$$

$$\frac{i}{2}(\overline{\psi}\gamma^\mu\nabla_\mu\psi - \nabla_\mu\overline{\psi}\gamma^\mu\psi) - XW_\sigma S^\sigma - m\Phi = 0, \tag{33}$$

$$\nabla^\mu S^\rho\varepsilon_{\mu\rho\alpha\nu} + i(\overline{\psi}\gamma_{[\alpha}\nabla_{\nu]}\psi - \nabla_{[\nu}\overline{\psi}\gamma_{\alpha]}\psi) + 2XW_{[\alpha}S_{\nu]} = 0; \tag{34}$$

$$i(\overline{\psi}\nabla^\alpha\psi - \nabla^\alpha\overline{\psi}\psi) - \nabla_\mu M^{\mu\alpha} - XW_\sigma M_{\mu\nu}\varepsilon^{\mu\nu\sigma\alpha} - 2mU^\alpha = 0, \tag{35}$$

$$(\nabla_\alpha\overline{\psi}\boldsymbol{\pi}\psi - \overline{\psi}\boldsymbol{\pi}\nabla_\alpha\psi) - \frac{1}{2}\nabla^\mu M^{\rho\sigma}\varepsilon_{\rho\sigma\mu\alpha} + 2XW^\mu M_{\mu\alpha} = 0; \tag{36}$$

$$\nabla_\alpha\Phi - 2(\overline{\psi}\sigma_{\mu\alpha}\nabla^\mu\psi - \nabla^\mu\overline{\psi}\sigma_{\mu\alpha}\psi) + 2X\Theta W_\alpha = 0, \tag{37}$$

$$\nabla_\nu\Theta - 2i(\overline{\psi}\sigma_{\mu\nu}\boldsymbol{\pi}\nabla^\mu\psi - \nabla^\mu\overline{\psi}\sigma_{\mu\nu}\boldsymbol{\pi}\psi) - 2X\Phi W_\nu + 2mS_\nu = 0; \tag{38}$$

which are the Gordon decompositions of Dirac equations. Among them, the best known (what in quantum field theory is called the Gordon decomposition in a strict sense) is (35).

Plugging into all of them the polar form of the spinor field and its covariant derivative allows us the obtention of

$$\nabla_\mu U^\mu = 0, \tag{39}$$

$$(\nabla_\mu\beta - 2XW_\mu + \tfrac{1}{2}\varepsilon_{\mu\alpha\nu\rho}R^{\alpha\nu\rho})U^\mu + 2P_\mu S^\mu = 0, \tag{40}$$

$$\nabla^{[\alpha}U^{\nu]} + \varepsilon^{\alpha\nu\mu\rho}(\nabla_\mu\beta - 2XW_\mu)U_\rho - \tfrac{1}{2}R^{ij}{}_\mu\varepsilon_{ij\rho\kappa}U^\kappa\varepsilon^{\alpha\nu\mu\rho} + 2\varepsilon^{\alpha\nu\mu\rho}P_\mu S_\rho - 2mM^{\alpha\nu} = 0; \tag{41}$$

$$\nabla_\mu S^\mu - 2m\Theta = 0, \tag{42}$$

$$(\nabla_\mu\beta - 2XW_\mu + \tfrac{1}{2}\varepsilon_{\mu\alpha\nu\rho}R^{\alpha\nu\rho})S^\mu + 2P_\mu U^\mu - 2m\Phi = 0, \tag{43}$$

$$\nabla^{[\alpha}S^{\nu]} + \varepsilon^{\alpha\nu\mu\rho}(\nabla_\mu\beta - 2XW_\mu)S_\rho - \tfrac{1}{2}R^{ij}{}_\mu\varepsilon_{ij\rho\kappa}S^\kappa\varepsilon^{\alpha\nu\mu\rho} + 2\varepsilon^{\alpha\nu\mu\rho}P_\mu U_\rho = 0; \tag{44}$$

$$\nabla_\mu M^{\mu\alpha} - 2XW_\sigma\Sigma^{\sigma\alpha} + \tfrac{1}{2}R^{ij\alpha}M_{ij} - 2P^\alpha\Phi + 2mU^\alpha = 0, \tag{45}$$

$$\nabla^\mu\Sigma_{\mu\alpha} + 2XW^\mu M_{\mu\alpha} + \tfrac{1}{2}R_{ij\alpha}\Sigma^{ij} + 2P_\alpha\Theta = 0; \tag{46}$$

$$\nabla_\alpha\Phi + (2XW_\alpha - \tfrac{1}{2}\varepsilon_{\alpha\mu\nu\rho}R^{\mu\nu\rho})\Theta + R_{\alpha\sigma}{}^\sigma\Phi + 2P^\mu M_{\mu\alpha} = 0, \tag{47}$$

$$\nabla_\nu\Theta - (2XW_\nu - \tfrac{1}{2}\varepsilon_{\nu\mu\sigma\rho}R^{\mu\sigma\rho})\Phi + R_{\nu\sigma}{}^\sigma\Theta - 2P^\mu\Sigma_{\mu\nu} + 2mS_\nu = 0; \tag{48}$$

which are the Gordon decompositions in the polar form. The equivalent of (35) is (45), and now we can see that this is precisely the Gordon decomposition in a strict sense of QFT because in the absense of torsion and for $R_{ij\alpha}$ equal to zero, it can be written as $P^\alpha\Phi = mU^\alpha + \nabla_\mu M^{\mu\alpha}/2$, telling us that $P^\alpha$ should be recognized as the momentum of the particles given in terms of kinematic momentum $mU^\alpha$ plus $\nabla_\mu M^{\mu\alpha}/2$ and therefore specifying that the angular momentum is the sum of the orbital angular momentum and the spin contribution [25].

The latter two, after a proper diagonalization, offer

$$\nabla_\mu \beta - 2XW_\mu + B_\mu - 2P^\iota u_{[\iota} s_{\mu]} + 2m s_\mu \cos \beta = 0, \tag{49}$$

$$\nabla_\mu \ln \phi^2 + R_\mu - 2P^\rho u^\nu s^\alpha \varepsilon_{\mu\rho\nu\alpha} + 2m s_\mu \sin \beta = 0, \tag{50}$$

specifying all derivatives of both degrees of freedom in terms of the trace $R_{va}{}^a = R_v$ and the Hodge dual $\varepsilon_{v\alpha\pi\iota} R^{\alpha\pi\iota}/2 = B_v$ as well as $P_\mu$ and the torsion of the space-time. By means of Identities (16) and (17), one can prove that these two equations imply Dirac Equation (28) in the polar form, and thus in general. Hence, (49) and (50) are equivalent to the Dirac equation [23].

Equations (49) and (50) can then be called Dirac equations in the polar form. They express the Dirac dynamics by means of the degrees of freedom $\phi^2$ and $\beta$ supplemented by $u_\alpha$ and $s_\alpha$, all of which have a clear meaning. Specifically, chiral angle $\beta$ is the phase difference between chiral parts: $s_\alpha$ is the spin, module squared $\phi^2$ is the density, $u_\alpha$ is the velocity of the matter distribution described by the spinor field. The necessary condition for the non-relativistic limit is to have $\beta = 0$ while the classical approximation is implemented by asking $s_\alpha \to 0$ [23]. As for density $\phi^2$ and velocity $u_\alpha$, they are exactly the density and velocity one would have in fluid mechanics. Thus, Equations (49) and (50) are what expresses the Dirac dynamics as a type of hydrodynamics, therefore extending to the relativistic case with spin the hydrodynamic formulation of quantum mechanics that was originally given by Madelung.

### 3. The 19 Formulations

We observed that the Dirac theory can be written in a hydrodynamic formulation. Our goal now is to find all ways in which this can be achieved, that is, find all manners to write a covariant set of field equations like (49) and (50).

To make things easier, we start by introducing the two vectors

$$2E_\mu = B_\mu - 2XW_\mu + \nabla_\mu \beta + 2m s_\mu \cos \beta, \tag{51}$$

$$2F_\mu = R_\mu + \nabla_\mu \ln \phi^2 + 2m s_\mu \sin \beta, \tag{52}$$

which, admittedly, is a definition that may appear arbitrary now, but which is essential to simplify all computations later. In fact, when in the full set of Gordon decompositions (39) and (48), we also substitute the bi-linear spinors in the polar form, using the above definitions. We obtain, respectively, the following sets:

$$F_\mu u^\mu = 0, \tag{53}$$

$$E_\mu u^\mu + P_\mu s^\mu = 0, \tag{54}$$

$$\varepsilon^{\alpha\nu\mu\rho} E_\mu u_\rho + F^{[\alpha} u^{\nu]} + \varepsilon^{\alpha\nu\mu\rho} P_\mu s_\rho = 0; \tag{55}$$

$$F_\mu s^\mu = 0, \tag{56}$$

$$E_\mu s^\mu + P_\mu u^\mu = 0, \tag{57}$$

$$\varepsilon^{\alpha\nu\mu\rho} E_\mu s_\rho + F^{[\alpha} s^{\nu]} + \varepsilon^{\alpha\nu\mu\rho} P_\mu u_\rho = 0; \tag{58}$$

$$F_\mu u_j s_k \varepsilon^{jk\mu\alpha} + E_\mu u^{[\mu} s^{\alpha]} - P^\alpha = 0, \tag{59}$$

$$F_\mu u^{[\mu} s^{\alpha]} - E_\mu u_j s_k \varepsilon^{jk\mu\alpha} = 0; \tag{60}$$

$$F_\mu - P^\rho u^\nu s^\alpha \varepsilon_{\mu\rho\nu\alpha} = 0, \tag{61}$$

$$E_\mu - P^\iota u_{[\iota} s_{\mu]} = 0; \tag{62}$$

as four groups of hydrodynamic field equations. The last group is, of course, (49) and (50), which we have already demonstrated to be equivalent to the Dirac equation. Therefore, any group that is proven to be equivalent to (61) and (62) is equivalent to the Dirac equation. In fact, to be more precise, since the Dirac equation is already implying all groups, all we need to achieve is prove that a group implies (61) and (62) to show the equivalence. This can be easily achieved with a formal tensor algebra. For instance, we take (59) and (60), and then, from (59), we isolate $P^\alpha$, while projecting from (60) along $u^\alpha$ and $s^\alpha$, obtaining

$$P^\alpha = F_i u_j s_k \varepsilon^{ijk\alpha} + E_i u^{[i} s^{\alpha]}, \tag{63}$$

$$E_\mu u_j s_k \varepsilon^{jk\mu\alpha} = 0, \tag{64}$$

$$F_\mu u^\mu = 0, \tag{65}$$

$$F_\mu s^\mu = 0 \tag{66}$$

to be used in (61) and (62). By substituting (63) and repeatedly using (64), (65) and (66), one can see that, in fact, (61) and (62) are verified. Therefore, (59) and (60) do imply (61) and (62). A similar proof can also be achieved for the first two groups. As a consequence, groups ((53)-(54)-(55)), ((56)-(57)-(58)), ((59)-(60)), ((61)-(62)) are equivalent to one another, and each to the Dirac equations, as it has already been discussed in [26]. These are already four ways in which the Dirac equations are written in a hydrodynamic form. Each one, consisting of exactly eight equations, is also the most stringent to be equivalent to the Dirac equation.

Nevertheless, we may drop the strict equivalence to find other sets of hydrodynamic forms equivalent to the Dirac equation up to redundancies. To achieve this, we choose the unitary gauge $u^0 = 1$ and $s^3 = 1$, obtaining, respectively for all groups, and in order within a single group, the following equations:

$$F_0 = 0, \tag{67}$$

$$E_0 + P_3 = 0, \tag{68}$$

$$F_1 + P_2 = 0, \quad F_2 - P_1 = 0, \quad F_3 = 0, \quad E_3 + P_0 = 0, \quad E_2 = 0, \quad E_1 = 0 \tag{69}$$

from the first group;

$$F_3 = 0, \tag{70}$$

$$E_3 + P_0 = 0, \tag{71}$$

$$E_2 = 0, \quad E_1 = 0, \quad F_0 = 0, \quad E_0 + P_3 = 0, \quad F_1 + P_2 = 0, \quad F_2 - P_1 = 0 \tag{72}$$

from the second group;

$$E_3 + P_0 = 0, \quad F_2 - P_1 = 0, \quad F_1 + P_2 = 0, \quad E_0 + P_3 = 0, \tag{73}$$

$$F_3 = 0, \quad E_2 = 0, \quad E_1 = 0, \quad F_0 = 0 \tag{74}$$

from the third group;

$$F_0 = 0, \quad F_1 + P_2 = 0, \quad F_2 - P_1 = 0, \quad F_3 = 0, \tag{75}$$

$$E_0 + P_3 = 0, \quad E_1 = 0, \quad E_2 = 0, \quad E_3 + P_0 = 0 \tag{76}$$

from the fourth group. As it is clear after some scrambling, all groups contain the same equations, proving again that they are all equivalent. However, now one can achieve more, because any grouping that would re-cover these eight equations would also obtain the validity of the Dirac equation in the unitary gauge, and hence in general. For example, we consider the equations given by (53), (56), (59), (62). They, respectively, yield (67), (70), (73), (76), which do recover all equations, with equations $E_0 + P_3 = 0$ and $E_3 + P_0 = 0$ repeated twice. Consequently, the set ((53), (56), (59), (62)) is equivalent to the Dirac equation with two redundant equations. Exactly the same can be said for the set ((54), (57), (60), (61)). Analogous reasonings would help us see that there are other sets of equations equivalent to the Dirac equation with a different number of redundant equations. All in all, we have the following list of sets:

1.  ((53), (56), (59), (62)), ((54), (57), (60), (61)) with two;
2.  ((53), (55), (59)), ((54), (55), (60)), ((54), (55), (61)), ((53), (55), (62)), ((56), (58), (59)), ((57), (58), (60)), ((57), (58), (61)), ((56), (58), (62)) with three;
3.  ((55), (58)) with four;
4.  ((55), (59), (61)), ((55), (60), (62)), ((58), (59), (61)), ((58), (60), (62)) with six;

all equivalent to the Dirac equation and where the last number indicates the number of redundant equations that each set has.

These 15 groups, added to the 4 groups seen before, amount to a total of 19 groups that are equivalent to the Dirac equation. And they are the smallest sets to be equivalent to the Dirac equation, in the sense that in none of them we can remove an equation without losing the equivalence to the Dirac equation in spite of the fact that there might be some component of a covariant equation that is redundant. For instance, in set ((53), (55), (59)), the equations given by $E_3 + P_0 = 0$, $F_2 - P_1 = 0$ and $F_1 + P_2 = 0$ are redundant, but removing them would mean removing Equation (59) and therefore also equation $E_0 + P_3 = 0$ which is not redundant, so that the equivalence to the Dirac equation will be lost. While all of the above sets are the smallest in this sense, this is not a property of any possible set. For example, the set given by the divergences and curls of velocity and spin ((53), (56), (55), (58)) is equivalent to the Dirac equation with six redundant equations, but two Equations (53) and (56) are fully redundant and they could be removed altogether, leaving a set that is still equivalent to the Dirac equation. The largest set would be produced if we consider all equations, obtaining a set with 24 redundant equations. There is, of course, no need to perform that, and hence the selection of the 19 smallest sets is a measure of the fact that the work was performed accurately enough to maintain a certain degree of simplicity.

The importance of selecting such sets can be judged by the fact that some of them might be remarkably easy to be interpreted. For instance, set ((55), (58)) consists of only curls of velocity and spin, and the curl is a concept with an easy visualization. If instead one is less at ease with the curl and more at ease with the velocity, it might be preferable to consider the divergence and curl of the velocity vector and take a system like ((53), (55), (59)) since Equation (59) is the one providing the momentum of the field [27], also easily visualizable. No matter the personal taste, all these sets are helpful for a visual interpretation of relativistic quantum mechanics as a special type of fluid, and, as a consequence of this fact, they are all necessary instruments for the investigation of various manifestly covariant spin-related properties of the de Broglie–Bohm version of quantum mechanics [27], so central in contemporary physics.

## 4. Conclusions

In this paper, we considered the general geometric construction of spinor fields re-formulated in the context of polar variables. After extending the formulation to all differential structures with the introduction of the tensorial connection, we re-wrote the Dirac differential field equations in a hydrodynamic form. In achieving this, we acquired the possibility to see that the Dirac equations are equivalent to the four groups ((53)-(54)-(55)), ((56)-(57)-(58)), ((59)-(60)), ((61)-(62)), but also to the eight groups ((53)-(55)-(59)), ((54)-(55)-(60)), ((54)-(55)-(61)), ((53)-(55)-(62)), ((56)-(58)-(59)), ((57)-(58)-(60)), ((57)-(58)-(61)), ((56)-(58)-(62)) up to three redundant compo-

nents of the field equations, to the four groups ((55)-(59)-(61)), ((55)-(60)-(62)), ((58)-(59)-(61)), ((58)-(60)-(62)) up to six redundant components, to the two groups ((53)-(56)-(59)-(62)), ((54)-(57)-(60)-(61)) up to two redundant components, and to the one group (55)-(58) up to four redundant components, for a total of nineteen groups of field equations constituting the minimal sets, in the sense that no covariant equation can be removed by any of them without losing equivalence.

All these 19 sets are the smallest to be equivalent to the Dirac equations, converting the quantum mechanics of the spinning relativistic particle into hydrodynamic form, the only form that is well-suited to render quantum mechanics visually interpretable, in the spirit that was first followed by de Broglie and Bohm. So what next?

There are two main objectives that would have to be achieved to bring this treatment to an acceptable level of completion, the first of which being the fact that writing everything in hydrodynamic form, treating the spinor field as one special type of fluid, does not in itself imply consistency. In fact, that specific type of fluid may still have problems, as pointed out in [28–30]. More has to be achieved to make sure that, seen as a special fluid, the spinor field is well defined.

The second problem is also a problem of the de Broglie–Bohm interpretation, and that is the fact that we still do not know how exactly we can treat multi-particle systems in relativistic contexts, and, more generally, we do not know how to make a second-quantized version of it. This problem has been open since the 1960s, and we are not trying to solve it here.

We hope, however, that with such a variety of formulations, at least some problems may be of easier solution in the future.

**Funding:** This work was carried out in the framework of activities of the INFN Research Project QGSKY and funded by Next Generation EU through the project "Geometrical and Topological effects on Quantum Matter (GeTOnQuaM)".

**Data Availability Statement:** The data presented in this study are available upon reasonable request from the corresponding author.

**Conflicts of Interest:** The author declares no conflict of interest.

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
