# Peer review of "Dirac Hydrodynamics in 19 Forms"

_symmetry, doi:10.3390/sym15091685_

Round 1

Reviewer 1 Report

Referee's report on the paper "Dirac hydrodynamics in 19 forms"
by L. Fabbri

The author discusses the formulation of the Dirac theory of a fermion particle
in terms of the tensor variables. The subject is of interest in view of the
fundamental importance of the Dirac equation, and the results obtained may
contribute to understanding of the spin dynamics. Accordingly, the paper can
be published in Symmetry journal, however, not in its present form. The present
manuscript is written in a rather sloppy way, and it contains quite some number
of unclear points.

1. The reference list appears to be incomplete. Without underestimating the
importance of refs. [1,2], the formulation of the Dirac theory in terms of
tensor variables was developed much earlier by E. T. Whittaker [Proc. Roy. Soc.
Lond. A158 (1937) 38], H. S. Ruse [Proc. Roy. Soc. Edinburgh 57 (1936-37) 97]
and by A. H. Taub [Ann. Math. 40 (1939) 937]. The book by V. A. Zhelnorovich,
Theory of Spinors and Its Application in Physics and Mechanics (Springer, 2019)
in this sense provides a useful overview of the subject.

2. I do not quite understand what is written on page 1 in Sec.1 Introduction.
Why the complex-valuedness of the Dirac matrices is considered as a "problem"?
For example, in electrodynamics the electric and magnetic fields, along with
the constitutive material tensors, are often viewed as complex-valued variables,
and this is not a problem at all. On the contrary, the use of the complex
analysis represents an advanced and convenient tool which helps to solve the
physical problems in a very efficient way. Comments are needed.

3. The author uses nonstandard notation which is rather confusing. I would
suggest to add a remark explaining that the matrix \pi = \gamma_5 . Actually
it is incomprehensible to me why one needs a new notation \pi instead of the
common \gamma_5. Convincing arguments are needed.

4. The author never explains what the matrix "L" is in (12)? My guess is that
this is a Lorenz spinor transformation, which therefore is determined by 6
independent real parameters. However, this obviously contradicts eq. (23),
since the right-hand side contains 7 = 1 + 6 independent components of the
scalar \zeta and the skew-symmetric tensor \zeta_{ij} = - \zeta_{ji}. Moreover,
it is unclear how the right-hand side of (23) can contain an electric charge q.
Does it mean that the matrix "L" is a function of q?

5. In sec. 3, eqs. (29)-(51) contain an object W which is described as an axial
vector of torsion. Does it mean that the connection in eqs. (21) and (22) also
contains the non-Riemannian torsion piece? Explanations are needed.

6. Eqs. (41)-(77) contain an object "P" which is described as the 4-momentum.
Explanations are needed what it is. How this object "P" is related to the spinor
wave function?

7. Eq. (66) is the same as (54), whereas eq. (67) coincides with (57). Why such
a duplication is needed?

8. On page 7, the author mentions the "unitary gauge" (in particular, on line
186, we read that "we may go in the unitary gauge"). What is it? How one can
"go into a unitary gauge"? Recalling that physics is an experimental science,
the author should be able to explain to experimenters how to "go into a
unitary gauge". Which experimental apparatus helps to do this?

9. Readers may wonder why (30)-(51) is a "hydrodynamic representation"? Which
objects can be interpreted as hydrodynamic variables and equations? How?

10. More explanations are needed in Sec. 4 concerning 19 formulations. I wonder,
in particular, are these formulations equivalent among themselves or not? If
they are, then the Abstract on page 1 is totally misleading, because on line 2
and 4, the author talks about "19 different sets" and "19 different ways". So,
are they different or equivalent?

11. In order to make the paper self-contained, the author should explain what
are "Fierz identities" and "Gordon decomposition"? Please explain: Identities
between which objects? Also: Decomposition of which objects are meant? Perhaps,
additional references to the corresponding literature sources would be helpful.

12. What is \Sigma^{ij} in eq. (8)? Please define this object.

Summarizing, a thorough revision is needed.

Minor polishing of English is needed.

Reviewer 2 Report

The major content of this paper is: first write out Dirac spinor in polar form,  then express Dirac equation in terms of some of the 8 observables (obtaining the so called hydrodynamic form of the Dirac equation), and further point out that these 8 observables are not independent, since they are subject to some constraint relations .  While the results are interesting and important, I must point out that these results have been derived fifty years ago by David Hestenes (Local observable in the Dirac theory, Jounal of Mathematical Physics 14, 893(1973)). Of course, Hestenes's work has not attracted much attention up to date, the reason is mainly due to the spacetime algebra he employed, since this mathematical language is not familiar to the most physicists. 

What I'm concerned most is some inappropriate or misleading  statements in the "Introduction", for example, "... the Clifford matrices come with three features that one may also see as problematic: one, these matrices are not uniquely defined, and hence different representations are possible; this apparent ambiguity is quenched by the fact that in 4-dimensions all representations are unitarily equivalent, although this cannot help us face the second point, and that is in general such matrices are complex-valued; ... ". The fact that Dirac's gamma matrices are not unique reflects the fact that there are infinitely many inertial frames which are different up to some Lorentz rotations (also infinitely many). The important thing is the basic algebra the four gamma matrices must satisfy, not their specific forms, although specific expressions are needed in actual calculations by convention.  So the non-unique representations needn't be "qunched"! The similar reasoning applies to two other "features". In summary, it is now well established that the hydrodynamic form of Dirac theory is equivalent to the standard form in the sense that both can successfully  predict the relevent experimental observations. Indeed, the two approaches distinguish one from the other by their different interpretations of quantum mechanics, such as particle trojectories, wave collaps, and so on, but not the features listed by the author.

Another example, " Because a relativistic spinor is complex-valued, each one of its components can be written as the product of a module times a unitary phase." The Dirac spinor is a 4X1 complex column matrix in its sdandard form, it has 8 degrees of freedom (8 real scalars), this is isomorphic to the polar form (also has 8 degrees of freedom, represented by 8 observables). So what's meaning of  "each one of its components ..."?

Reviewer 3 Report

This paper is enlightening,well written and useful.  I recommend publication but some issues need to be clarified first.

a) The author should define clearly what they mean by "hydrodynamics" here.  In Bohmian mechanics, hydrodynamics means the equation of continuity for probability, energy and momentum, which is enough to give a Schrodinger-equation type dynamics.  But as recently has been found (see the recent literature on relativistic hydrodynamics with spin, some authors to look for are Montenegro,Florkowski,Rischke) one cannot easily extend this form of hydrodynamics to "ideal" fluids with spin, because one cannot get enough local conservation equations.

b) In the introduction the author says all representations of Dirac matrices are unitarily equivalent.   I can not see how this can be true for Dirac and Majorana spinors.    Please clarify if the fluid picture can be also defined for Majorana fermions

c)  This discussion seems to be appropriate for the non-interacting Dirac equation as quantum mechanics (with its associated paradoxes), not for quantum field theory.  The author should comment on this.  Note that general interactions added via bilinears can be complex, this is what happens with CP violating Yukawa terms.

Round 2

Reviewer 1 Report

Referee's report on the revised paper "Dirac hydrodynamics in 19 forms"
by L. Fabbri

The author has answered satisfactorily to the questions of my first report.
The details of the formalism are now included, literature updated, notations
and computations are explained in an appropriate way. As a result, the paper
is in a better shape, in particular, it is more readable now. Therefore
I recommend a publication of the revised paper in its present form.

Minor polishing of English style and spelling may be needed.

Reviewer 3 Report

The author answered my questions satisfactorily, I recommend publication